# Proposal to Consider Chemical/Physical Microenvironment as a New Therapeutic Off-Target Approach

**DOI:** 10.3390/pharmaceutics14102084

**Published:** 2022-09-29

**Authors:** Alessandro Giuliani, Stefano Fais

**Affiliations:** 1Department of Environment and Health, Istituto Superiore di Sanità, 00161 Rome, Italy; 2Department of Oncology and Molecular Medicine, Istituto Superiore di Sanità, 00161 Rome, Italy

**Keywords:** drugs, off-targeting, microenvironment, proton pump inhibitors, proton exchangers inhibitors, diseases

## Abstract

The molecular revolution could lead drug discovery from chance observation to the rational design of new classes of drugs that could simultaneously be more effective and less toxic. Unfortunately, we are witnessing some failure in this sense, and the causes of the crisis involve a wide range of epistemological and scientific aspects. In pharmacology, one key point is the crisis of the paradigm the “magic bullet”, which is to design therapies based on specific molecular targets. Drug repurposing is one of the proposed ways out of the crisis and is based on the off-target effects of known drugs. Here, we propose the microenvironment as the ideal place to direct the off-targeting of known drugs. While it has been extensively investigated in tumors, the generation of a harsh microenvironment is also a phenotype of the vast majority of chronic diseases. The hostile microenvironment, on the one hand, reduces the efficacy of both chemical and biological drugs; on the other hand, it dictates a sort of “Darwinian” selection of those cells armed to survive in such hostile conditions. This opens the way to the consideration of the microenvironment as a convenient target for pharmacological action, with a clear example in proton pump inhibitors.

## 1. Introduction

Notwithstanding the ever-increasing expenditure in pharmacological research and the claimed ever-increasing and detailed knowledge of biological mechanisms, the number of newly marketed drugs and the ratio between new drugs and expenditure have been rapidly falling since the 1980s [1]. Moreover, despite the promises of a “druggable genome” set forth by the rise of the genomic (and post-genomic) era, the majority of newly marketed drugs interact with “old” receptors discovered well before this era [2].

The crisis encompasses a wide range of epistemological issues related to the general crisis of reductionism in science. This crisis, in the case of drug discovery, has its focus on the paradigm of a “magic bullet” capable of interfering with a supposed “critical step” of a largely deterministic chain of events that goes from the molecular to the organismic layer of organization [3].

The feasibility of drug repurposing stems from the presence of a huge “dark side of pharmacology” that involves the off-target (and, in general, unexpected) effects of known drugs coming from the presence of intermingled interaction networks preventing any effort of developing drug molecules endowed with a single molecular mechanism of action [4,5].

In the first part of this article, we briefly set out why a strategy rooted in the quest for the “correct receptor to hit” cannot work in the case of intermingled interaction networks [6], with a particular emphasis on cancer [7,8]. In this section, we outline the essentials of the complex network paradigm and the consequent acquiring of a conscious approach in facing complexity [9]. The network paradigm indicates the emergence of “bio-dynamic interfaces” [10] as the only way to mediate the interaction between complex systems, such as the organism and its environment, at all layers of biological organization. In contrast, the network pharmacology approach is strictly linked to rational drug re-purposing [11] by promoting a process-focused disease description over the usual phenotype-based one.

In the second section of the article, the prevailing role of context (microenvironmental cues and physical forces) with respect to receptor-based mechanistic rules is demonstrated as one of the main drivers of cancer treatment failure [12,13]. Other relevant issues, such as the dependence of the microenvironment on the successes and failures of antibiotic therapies [14] and in the management of gut functionality [15], are also outlined.

The last part of the review is devoted to the analysis of diverse successful cases of drug repurposing spanning different application fields, with a particular emphasis on non-canonical pharmacodynamics [16,17] and newly proposed computational hypothesis testing approaches [18,19,20].

The term microenvironment indicates the entire context that the biological system is involved in, even the chemical/physical forces impinging on (and, in turn, being modified by) the system, and throughout this work we concentrate on these forces. This choice was dictated by both the impossibility of exhaustively taking into consideration all the intricacies of microenvironment factors and of highlighting a still largely unexplored avenue of drug action different from the usual molecular organization layer taken into consideration by pharmacological research.

## 2. The Network Paradigm

The multi-level organization of nature is self-evident: at the very basic level of biological organization, proteins interact among themselves to give rise to an organized metabolism, while, at the same time, each protein (a single node of such an interaction network) is itself a network of interacting amino acid residues. Similar pictures can be drawn for the structure and function of cells, organs, tissues, and ecological systems. The network paradigm indicates a circular causation pattern in which bottom-up (the perturbation of more basic layers influences higher level layers) and top-down (the global architecture of the network impinges on the behavior of a single node) causation models are both relevant for the same phenomenon.

This evidence is in sharp contrast to the usual way of approaching pharmacology by an exclusive bottom-up quasi-deterministic approach: causally relevant events originate from the lower level (the molecular one) in the form of perturbations that “climb up” the hierarchy, reaching the ultimate layer of macroscopic behavior (e.g., causing a specific disease) [21]. The rising interest in complex network studies allows scientists to further the pure qualitative observation of the existence of both non-linear and non-bottom-up processes, and to uncover the deep nature of multi-level organization. As pointed out by Nicosia et al. [22], “Networks are the fabric of complex systems”. Only a network organization, essentially consisting of a set of nodes (elementary players) linked by edges (mutual interactions), can give rise to the classical attributes of living matter: the presence of multiple equilibrium states, robustness, and the possibility to adapt to an ever-changing environment [23]. In a network system, the single players do not work in isolation: the “passage of scale” from the microscopic level (e.g., single node perturbation) to the global response of the system (e.g., the organism outcome) emerges from the re-arrangement of the entire network in which the activities of the single nodes are influenced by local perturbation. In the majority of cases, this perturbation is buffered by means of the rapid extinction of the initial stimulus [23], which dissipates while spreading across the network. This phenomenon is both the basis of biological system resilience and of the missed promise of a “druggable genome”. Being only a strict minority and, in most cases, already “drugged” in past decades [2], the nodes whose stimulation provokes a persistent perturbation are able to initiate a regime shift of the entire system. The recognition of this fact fostered two new pharmacological lines of research: an approach mimicking allostery, the most evident “global reorganization” of a network system by an effective stimulus [24], and the search for “network drugs” that, instead of targeting a single molecular entity, act as “weak binders” at multiple sites of the interaction network [25]. 

Biological networks are embedded into an ever-changing microenvironment that, in turn, can be considered a network of mutual interactions among its constituents. This embedding is far from being a uni-directional relationship. Rather, it is a continuous bi-directional accommodation of the microenvironment. In these “microcosmo” cells, vesicles, soluble factors, vessels, and chemical and physical factors crosstalk continuously in an unceasing modification of the biological system. This implies the existence of a “field” encompassing both the biological system and its microenvironment. The concept of a “field” is crucial to catch the essence of considering a microenvironment as a target for drug action. From basic physics, we know that a point charge embedded into an electromagnetic field both “senses” (i.e., is influenced by the field) and modifies (i.e., influences) the field. This is exactly what happens with a microenvironment, in which environmental cues influence the biological system, and, consequently, the biological system modifies its microenvironment (e.g., pH modifications triggered by the Warburg effect and the continuous acidification supported by the H+ extracellular elimination by proton pumps). This mutual interaction is made possible by a shared biodynamic interface that changes in both time and space [10].

## 3. Microenvironment and Diseases

The role of the microenvironment in disease pathogenesis and, most of all, in response to therapies based on drugs is sadly neglected. An example, among many, is cancer. While cancer is commonly described as “a disease of the genes”, it is also associated with massive metabolic re-programming, which is now accepted as a disease hallmark [26,27]. This programming is complex and often involves metabolic cooperativity between cancer cells and their surrounding stroma. Indeed, there is emerging clinical evidence that interrupting a cancer’s metabolic program can improve patients’ outcomes [28,29,30]. The most commonly observed and well-studied metabolic adaptation in cancers is the fermentation of glucose to lactic acid, even in the presence of oxygen, also known as “aerobic glycolysis” or the “Warburg effect”. Much has been written about the mechanisms of the Warburg effect, and this remains a topic of great debate [27,31]. Independently from the mechanism underlying this phenomenon, the ultimate outcome of aerobic glycolysis is the acidification of the tumor microenvironment. Rather than being an epiphenomenon, it is now appreciated that this acidosis is a key player in cancer somatic evolution and progression to malignancy [26,32]. Adaptation to acidosis induces and selects malignant behaviors, such as increased invasion and metastasis, chemoresistance, and the inhibition of immune surveillance. However, the metabolic reprogramming that occurs during the adaptation to acidosis also introduces therapeutic vulnerabilities. Thus, tumor acidosis is a relevant therapeutic target, and there are some reasonable approaches for accomplishing this: (1) neutralizing acid directly with buffers, (2) targeting the metabolic vulnerabilities revealed by acidosis, (3) developing either acid-activatable drugs or acid-sensible nanocarriers, and (4) inhibiting the metabolic processes responsible for generating acids. Several cellular functions are dictated by the pH differences between extracellular and intracellular spaces. These differences are called “pH gradients”, and they are, for instance, crucial for the uptake of chemical drugs by target cells. Figure 1 illustrates a rudimentary portrait of this phenomenon.

However, some strategies have been used to exploit these potentially very powerful approaches [26,33]. These strategies include (i) the use of simple buffers, such as sodium bicarbonate; (ii) more complex buffers, such as some mixes of carbonates and bicarbonates (e.g., Basenpulver); and (iii) the use of a series of inhibitors of ion/proton exchangers. The vast majority of the studies involving either buffers or exchange inhibitors are confined to pre-clinical studies, but proton pump inhibitors (PPIs), which are being extensively used worldwide as anti-acidic drugs, have been combined with chemotherapy in some clinical trials. The results of both pre-clinical and clinical studies have convinced scientists to think about the re-purposing of these classes of drugs as anti-tumor drugs [33,34].

Epidemiological studies have shown that numerous risk factors are shared by diabetes and several cancer sites. Among these primarily are obesity and smoking status, but they also include low physical activity and alcohol consumption. The pathophysiological mechanisms implicated in the association between Type 2 Diabetes (T2D) and cancer have been proposed for colorectal, pancreas, and liver cancers. These include the T2D microenvironment, as represented by advanced glycation end-products; chronic local inflammation; hyperlipidemia; extracellular matrix disorders; and altered microbiota that could predispose the development of colorectal cancer. However, despite the strong epidemiological evidence, the mechanistic bases of the association between diabetes and cancer are still not understood [35].

The major sub-type of T2D is peripheral insulin resistance associated with obesity and central adiposity, leading to hyperinsulinemia and chronic inflammation, both of which have the potential to exacerbate the risk of cancer. Hyperinsulinemia, together with hyperglycemia, also contributes to the accumulation of keto acids, leading to chronic systemic acidosis, which is compensated for by reducing HCO3 and by reducing the metabolic interstitial buffering capacity, making interstitial pH more fragile. Hyperinsulinemia is also associated with increased circulating levels of insulin-like growth factor-1 (IGF1), which is a potent mitogenic factor for neoplastic epithelial cells. The binding of IGF1 to its receptor triggers the activation of the PI3K➔Akt➔mTOR pathway, inducing metabolic activation and mitogenesis [35].

What cancer and diabetes have in common is the increased reliance on glucose fermentation. Continuous glucose fermentation leads to lactate production and significant local acidosis in both diabetic peripheral tissues and tumors. Acidosis is exacerbated if combined with decreased perfusion, which can be a consequence of inflammation, peripheral vascular resistance, and dysangiogenesis, all common syndromes in cancer and diabetes. There is significant evidence, presented below, that this local acidosis in cancer can promote tissue re-modeling, local invasion, metastasis, and the inhibition of immune surveillance. In diabetes, local and systemic acidosis reduces insulin’s affinity for its receptor, exacerbating the spiral of peripheral insulin resistance. Consequently, targeting acidosis is an important therapeutic approach in both T2D and cancer, as discussed below.

The same principles hold true for another major health treatment: the rising resistance to antibiotic therapy. In [14], the authors explicitly take into account the consideration of a microenvironment as a target of antibiotic treatment in terms of the modification of microenvironmental cues so as to inhibit the emergence of biofilms, organized colonies of bacterial cells, which are very difficult to eradicate with antibiotics. It is worth noting that antibiotic efficacy is routinely assessed by antibiograms, in which the sensitivity of the bacterial strains responsible for an infection in a given patient is tested in vitro. The puzzle resides in the fact that bacteria behave substantially differently in standard laboratory conditions from actual infections. The infectious microenvironment imposes changes in growth and metabolic activity that result in increased protection against antibiotics (e.g., biofilm production). Therefore, an improved antibiotic treatment of chronic infections is achievable when antibiotics are recommended based on susceptibility testing in relevant in vitro conditions that resemble actual infectious microenvironments, with a particular focus on fostering direct interventions on the microenvironment that could prevent biofilm formation.

Along a similar vein (after all, we are dealing with ecological systems, and, again, we can safely consider cancer as an ecological threat with a species going out of control), we can posit direct intervention on the microenvironment of microbiota [15]. The microbiome can be considered a complex biocenosis stemming from a network of interactions between thousands of bacterial and yeast species. It is not by chance that microbiome research is based on the same computational and theoretical principles as ecology [36]. The microbiome is involved in the creation of complex biodynamic interfaces with host tissues, e.g., the ability to colonize and thrive within the mucous layer that covers the colon epithelium. These mucosal microbes intimately interact with the intestinal tissue and are important modulators of human health. Embedded in the host-secreted mucous matrix, they form a “mucosal biofilm” with a distinct composition and functionality that, in turn, shapes the mucosal microenvironment. This implies the need to actively consider the mucosal microenvironment as a target for any therapeutic intervention for re-establishing the correct microbial ecology.

## 4. Rethinking off-Targeting for Treating Microenvironment—An Eco-evolutionary Way of Thinking

In dealing with microenvironments, tumors represent an emblematic issue. As introduced above, tumors survive in extreme conditions due to the upregulation of a series of proton extrusion pumps [34], which release protons and lactate into the extracellular environment; this avoids the acidification of the cytosol, which inevitably kills any cells. Among the proton flux regulators are vacuolar H+-ATPases (V-ATPases), Na+/H+ exchangers (NHE), monocarboxylate transporters (MCTs), carbonic anhydrase IX (CA-IX), and Na+/HCO3 co-transporters (NBCs) [34]. However, it is a common belief that this phenomenon is not the result of transformation but rather the result of “cell clone selection”. In fact, uncontrolled growth, lactic and carbonic acid production, and low blood and nutrient supply contribute to the generation of a tumor microenvironment that is extremely toxic for either normal or more differentiated cells, thus progressively selecting cells capable of surviving in these adverse conditions. This phenomenon occurs independently from the tumor histotype.

Notably, normal cells at a pH ranging from very acidic to weakly acidic inevitably die or are entirely blocked in their functions [35].

For these reasons, a therapy based on a combination of existing proton or ion efflux pump inhibitors is under investigation in order to determine whether the implementation of molecules targeting several mechanisms underlying tumor acidification may have a significant effect on tumors. To do this, a series of drugs, whose main target is not tumors, have been considered. This list includes (i) proton pump inhibitors (Lansoprazole, Omeprazole, Esomeprazole, Rabeprazole, and Pantoprazole), whose main purpose is to act as an anti-acidic treatment for gastroprotection; (ii) carbonic anhydrase inhibitors (Acetazolamide), whose main target is glaucoma; (iii) inhibitors of Na+/H+ exchangers, including Amiloride, whose main purpose is to act as anti-hypertensive treatment for their diuretic activity, and Cariporide, whose main target is myocardial ischemia; (iv) monocarboxylate transporters inhibitors with multiple indications, such as the prevention of cardiac graft rejection; and (v) Na+/HCO3 co-transporters inhibitors thought to counteract heart failure.

However, at least as far as tumors are concerned, there is a general hypothesis supporting the progressive establishment of a sort of clonal cell selection induced by the setting of a very hostile microenvironment. This has also been called the “eco-evolutionary theory” of tumors, and it is considered a major actor in dictating the progressive isolation of growing tumors from the rest of the body [37], as well as in leading to an unresponsiveness to therapies [38]. One piece of evidence supporting this hypothesis is the discovery of cannibalism and, more generally, the “cell-in-cell” phenomena in cancers [39], witnessing a sort of primeval behavior in malignant tumor cells. It must also be considered that cannibalism is an almost exclusive phenotype of cells deriving from metastatic lesions [40]. This introduces a new concept in cancer therapy: to cure the microenvironment by re-tuning the eco-evolutionary selection of very malignant cells.

In a similar vein, the targeting of microenvironments has been invoked to deal with bacterial biofilms [41]. Inorganic nanoparticles with intrinsic antibacterial activity and inert nanoparticles assisted by external stimuli, including heat, photons, magnetism, or sound, have been found to be efficient against persistent infections [42]. These strategies are explicitly designed to target the unique microenvironment of bacterial infections.

Off-target experimentation is strictly connected to the consideration of a microenvironment as a proper therapeutic “target”. Here, we wish to stress the fact that this “target” is substantially different to a receptor molecular entity (e.g., there is no specific receptor to modify pH) and requires the abandoning of a strict reductionist approach to pharmacology.

## 5. Discussion

Here, we wish to propose a new paradigm in pharmacology that puts together the following well-established and recognized issues: (i) many known drugs have off-targets that very often may be recognized by their side effects [40]; (ii) changes in the extracellular microenvironment may represent one phenotype common to many diseases [34]; (iii) some features of the microenvironment may well represent a target for some known drugs, as has been shown for PPIs [34]; and (iv) the example of proton pump inhibitors should be taken into careful consideration when designing new drugs. In fact, as has been extensively shown for tumors, the vast majority of drugs once ending in a H+-enriched microenvironment are protonated and blocked outside tumor cells. PPIs are administered as prodrugs that are transformed in their active molecule (*tetracyclic sulphenamide)* only following protonation, and both pre-clinical and clinical evidence suggests that they are extremely active in the tumor microenvironment [34]. In fact, both pre-clinical and clinical investigations have shown that pre-treatment with PPIs allows other drugs, which are mostly ineffective in single therapies, to work [43].

To further emphasize the importance of one of the major microenvironmental phenotypes (i.e., extracellular acidosis) in conditioning the efficacy of past and new therapies, it has been shown that the same cell lines cultured at either a “physiological” (7,4) or acidic (6,5) pH display a very different lipid composition of the cell’s plasma membrane [44], which, together with increasing exosome release [12], reducing chemical drug efficacy [45], and inhibiting immune reaction against tumors [46], make tumor microenvironmental acidity a major determinant in anti-tumor therapy failure.

However, as is clearly emphasized in this article, a harsh microenvironment is a major determinant in the pathogenesis of many diseases [35,47], strongly supporting a new era in which the microenvironment may well represent a major target for new therapeutic molecules, while also considering the off-targeting of old drugs. Scientific evidence supports the role of acidity in impairing the immune response. In fact, systemic buffering improves the immune reaction against tumors [48]. However, the solid literature supports, either directly or indirectly, a key role of microenvironmental acidity not only in tumors but also in a series of pathological conditions involving metabolism, the cardiovascular system, the nervous system, infectious diseases, inflammation and the immune system, renal function, pain, and other pathological conditions. Table 1 summarizes these data with the appropriate references. However, some reviews have emphasized the importance of acidity in tumors, proposing some approaches to counteract it [26,27,49]. Many of the purposes behind newly treated microenvironmental acidity come from the new paradigm of using the off-targeting of old drugs [5], but some interesting and novel approaches to finding new anti-cancer drugs in the panel of available drugs have recently been proposed [50], thus opening a promising future for this issue. Lastly, two recent papers have shown that an anti-acidic treatment may reduce the risk of lung cancer [51] and that urine acidity is associated with poor prognosis in advanced bladder cancers [52], further suggesting that targeting microenvironmental acidity may represent a future strategy in both the prevention and treatment of diseases.

## Figures and Tables

**Figure 1 pharmaceutics-14-02084-f001:**
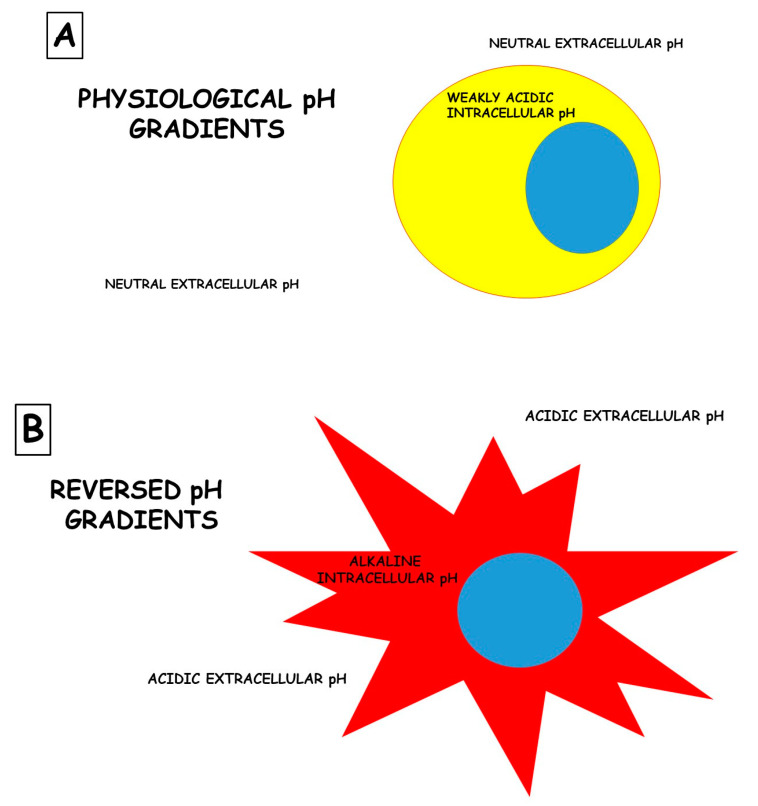
Differences in the pH gradients between normal (**A**) and tumor (**B**) cells. Notably, the presence of a deranged pH gradient in tumor cells does not allow chemical drugs to enter the cells and carry out their effects; instead, they are entirely protonated and, therefore, blocked outside the cells.

**Table 1 pharmaceutics-14-02084-t001:** pH and diseases. Direct and indirect evidence that microenvironmental acidity is involved in human diseases.

Disease	Histology	Type of Evidence	References
**Tumor**	Melanoma	Pre-clinical	[45]
		Pre-clinical	[53]
		Pre-clinical	[48]
		Pre-clinical	[54]
		Pre-clinical	[55]
		Pre-clinical	[56]
		Pre-clinical	[57]
		Pre-clinical	[58]
		Pre-clinical	[59]
	Lymphoma	Pre-clinical	[60]
		Pre-clinical	[61]
		Clinical	[62]
		Clinical	[63]
	Myeloma	Pre-clinical	[64]
		Pre-clinical	[65]
	Colon	Pre-clinical	[45]
		Pre-clinical	[66]
		Pre-clinical	[67]
	Breast	Pre-clinical	[68]
		Pre-clinical	[69]
		Pre-clinical	[70]
		Pre-clinical	[71]
		Pre-clinical	[72]
		Pre-clinical	[73]
		Pre-clinical	[74]
		Pre-clinical	[75]
		Clinical	[29]
	Ovary	Pre-clinical	[45]
		Pre-clinical	[76]
	Pancreas	Pre-clinical	[70]
		Pre-clinical	[77]
		Clinical	[78]
	Gastrointestinal	Pre-clinical	[79]
		Pre-clinical	[80]
		Pre-clinical	[81]
		Pre-clinical	[82]
		Pre-clinical	[83]
		Pre-clinical	[84]
		Pre-clinical	[85]
		Clinical	[30]
	Prostate	Pre-clinical	[86]
		Pre-clinical	[87]
	Liver	Pre-clinical	[88]
		Pre-clinical	[89]
		Pre-clinical	[90]
		Pre-clinical	[91]
		Clinical	[92]
	Lung	Clinical	[62]
		Clinical	[63]
	Osteosarcoma	Clinical	[28]
	Sarcomas	Pre-clinical	[93]
		Clinical	[62]
		Clinical	[63]
	Nervous system	Pre-clinical	[94]
		Pre-clinical	[95]
	Sensitivity to radiation therapy	Pre-clinical	[96]
		Pre-clinical	[97]
**Cardiovascular diseases**		Pre-clinical	[98]
		Clinical	[99]
		Clinical	[100]
		Clinical	[101]
**Metabolic diseases**		Pre-clinical	[102]
		Pre-clinical	[103]
		Pre-clinical	[104]
		Pre-clinical	[105]
		Pre-clinical	[106]
		Pre-clinical	[107]
		Clinical	[108]
[101]			
**Nervous system**		Pre-clinical	[109]
		Pre-clinical	[110]
		Pre-clinical	[111]
		Pre-clinical	[112]
		Pre-clinical	[113]
		Pre-clinical	[114]
**Inflammation and Immune system**		Pre-clinical	[115]
		Pre-clinical	[116]
		Pre-clinical	[117]
		Pre-clinical	[118]
		Pre-clinical	[119]
		Pre-clinical	[120]
**Renal function**		Clinical	[121]
**Pain**		Clinical	[122]
		Clinical	[123]
		Clinical	[124]
**Infectious** **agents**	Viruses	Preclinical	[125]
		Preclinical	[126]
		Preclinical	[127]
		Preclinical	[128]
		Translational	[129]
	Candida and other yeasts	Preclinical	[130]
		Preclinical	[131]
		Preclinical	[130]
		Preclinical	[132]
	Bacteria	Preclinical	[133]
		Preclinical	[134]
		Preclinical	[135]
		Preclinical	[136]
		Preclinical	[137]
		Preclinical	[138]
		Preclinical	[139]
	Parasites	Preclinical	[140]
		Preclinical	[141]
		Preclinical	[142]
		Preclinical	[143]
**Other conditions**	Pre-eclampsia	Translational	[144]
		Translational	[145]
	Genetic diseases	Preclinical	[146]

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
