# Peer review of "Proposal to Consider Chemical/Physical Microenvironment as a New Therapeutic Off-Target Approach"

_pharmaceutics, 2022, doi:10.3390/pharmaceutics14102084_

Round 1
Reviewer 1 Report
Manuscript ID: pharmaceutics-1870903.
Manuscript type: Perspective.
May We Consider Microenvironment As A New Therapeutic Off-Target
Alessandro Giuliani, Stefano Fais
Giuliani A. et al. discussed in this perspective manuscript the relevance of the sometimes forgotten microenvironment for therapeutic purposes in human diseases. The manuscript covers a significant subject that deserves to be considered when performing biomedical research and for ensuring a directed and successful therapy in affected patients. There are some concerns that must be solved first before the acceptance of the manuscript.
a) Major comments
1. The first thing that was interesting to me was finding that this manuscript is considered as a perspective. By looking at the online site of Pharmaceutics (https://www.mdpi.com/journal/pharmaceutics/instructions) I could not see this type of publication. Thus, as a reviewer I am unaware of the requirements for this type of manuscripts in particular, so I will consider it as a short review. Besides, the same authors (line 63) wrote this was indeed a review. In this sense, it is required to add at least one figure to the manuscript. An interesting idea could be to summarize the main topics covered here, such as cancer, diabetes and bacterial infections. The authors could represent the microenvironment in each one of these conditions, and could exemplify how they suggest to target the components in their respective microenvironments. In a second figure, they could show the readers the relationship between cancer and diabetes, as they suggest from lines 155 to 174.
2. The authors cover mainly changes in the pH and in the metabolism of the microenvironment. However, it is requested to discuss two additional topics as well: the effects of the intercellular communication through exosomes and paracrine stimuli, and importantly, how the immune cells infiltrated in the microenvironment could modify the response to therapy. The authors are invited to discuss, among others, 10.1007/s10555-013-9441-9, 10.1016/j.bbcan.2019.04.004, 10.1016/j.ebiom.2021.103627, 10.1186/s12885-021-09151-2, 10.1016/j.isci.2022.103973, 10.1186/s12964-020-00605-x, and 10.1371/journal.pone.0212275.
3. Even though the English of the manuscript is overall understandable, it is clear that it needs the hiring of a professional English editing service. There are several typos thorough the text and extra spaces between some words, and some words such as fungine (line 193) and Pantpprazole (line 224) seem confusing. The authors continuously employ capital letters when referring to compounds (i.e., Acetazolamide, Cariporide, Omeprazole, etc.) and even when referring to drug classifications (i.e., Carbonic Anhydrase Inhibitors, Inhibitors of Na+/H+ exchangers, etc.), which is not correct. Sometimes they even change the way they write some words, such as “micro environment” (line 58) and “micro-environment” (line 61). Therefore, the authors must show the certificate of a professional English editing service when resubmitting their manuscript for the next round of evaluation.
b) Minor comments
4. The whole phrase from lines 223 to 231 seems confusing due to the way it is written. The authors are only listing compounds versus the tumor microenvironment in cancer, but the reading is complicated. Probably after the English editing service this phrase would become more understandable.
5. Could the authors please deepen into how “tumors are concerned”? This phrase is seen in line 232.
6. Additionally, please illustrate in the cannibalism of cancer cells (line 237). This is a topic not commonly seen in cancer research.
Author Response
REVIEWER 1
Giuliani A. et al. discussed in this perspective manuscript the relevance of the sometimes forgotten microenvironment for therapeutic purposes in human diseases. The manuscript covers a significant subject that deserves to be considered when performing biomedical research and for ensuring a directed and successful therapy in affected patients. There are some concerns that must be solved first before the acceptance of the manuscript.
- a) Major comments
- The first thing that was interesting to me was finding that this manuscript is considered as a perspective. By looking at the online site of Pharmaceutics (https://www.mdpi.com/journal/pharmaceutics/instructions) I could not see this type of publication. Thus, as a reviewer I am unaware of the requirements for this type of manuscripts in particular, so I will consider it as a short review. Besides, the same authors (line 63) wrote this was indeed a review.
In this sense, it is required to add at least one figure to the manuscript. An interesting idea could be to summarize the main topics covered here, such as cancer, diabetes and bacterial infections. The authors could represent the microenvironment in each one of these conditions, and could exemplify how they suggest to target the components in their respective microenvironments. In a second figure, they could show the readers the relationship between cancer and diabetes, as they suggest from lines 155 to 174.
Actually, a Perspective article is provided by Pharmaceutics and more in general in all the MDPI journal, and to be honest our article has been written in this sense, thus emphasizing chemico-physical microenvironment as a ‘target’ for drug development. For these reason we don’t move from our initial idea. As an example the article below explicitly reports the term ‘Perspective’:
Open AccessPerspective
Working within the Design Space: Do Our Static Process Characterization Methods Suffice? by Moritz von Stosch et al. Pharmaceutics. 2020 Jun 17;12(6):562. doi: 10.3390/pharmaceutics12060562.
- The authors cover mainly changes in the pH and in the metabolism of the microenvironment. However, it is requested to discuss two additional topics as well: the effects of the intercellular communication through exosomes and paracrine stimuli, and importantly, how the immune cells infiltrated in the microenvironment could modify the response to therapy. The authors are invited to discuss, among others, 10.1007/s10555-013-9441-9, 10.1016/j.bbcan.2019.04.004, 10.1016/j.ebiom.2021.103627, 10.1186/s12885-021-09151-2, 10.1016/j.isci.2022.103973, 10.1186/s12964-020-00605-x, and 10.1371/journal.pone.0212275.
While we agree with the reviewer on the importance of the additional items suggested, again we must remark that this is not a review paper, in order to avoid misunderstandings we changed the title into ‘Can chemico-physical microenvironment..’. This limitation to ‘chemico-physical microenvironment’ emphasizes the basic concept of ‘microenvironment as a field’. We can think at an electric field that impinges on an embedded charged particle that in turn modifies the field, by substituting charged particle with cell (tissue) and electric field as the combination of chemical species and physical forces surrounding it. This minimalistic definition is, at least in our opinion, very important in order not to make the message too vague. In our opinion, this is essentially the novelty of the proposal: while do exist many papers dealing with angiogenesis modulation and immune therapies, the theme of chemico-physical modification of microenvironment is largely neglected.
- Even though the English of the manuscript is overall understandable, it is clear that it needs the hiring of a professional English editing service. There are several typos thorough the text and extra spaces between some words, and some words such as fungine (line 193) and Pantpprazole (line 224) seem confusing. The authors continuously employ capital letters when referring to compounds (i.e., Acetazolamide, Cariporide, Omeprazole, etc.) and even when referring to drug classifications (i.e., Carbonic Anhydrase Inhibitors, Inhibitors of Na+/H+ exchangers, etc.), which is not correct. Sometimes they even change the way they write some words, such as “micro environment” (line 58) and “micro-environment” (line 61). Therefore, the authors must show the certificate of a professional English editing service when resubmitting their manuscript for the next round of evaluation.
We thank the reviewer for the suggestion and, by the aid of a native speaker, we changed the text.
- b) Minor comments
- The whole phrase from lines 223 to 231 seems confusing due to the way it is written. The authors are only listing compounds versus the tumor microenvironment in cancer, but the reading is complicated. Probably after the English editing service this phrase would become more understandable.
- Could the authors please deepen into how “tumors are concerned”? This phrase is seen in line 232.
- Additionally, please illustrate in the cannibalism of cancer cells (line 237). This is a topic not commonly seen in cancer research.
We have done the requested changes, however as far as the cannibalism is concerned the it is a long story and the readers can well find all the details in the published papers, most of all in The Fais and Overholtzer Nature Reviews in cancer 2018
Reviewer 2 Report
The authors uncovered microenvironment as a convenient target for pharmacological action Major:
1) The rationale of why the authors came up with this review.
2) What is the information that is not exactly available that motivated the authors to come up with this information.
3)What are the current caveats and how do the authors highlight the current research in answering them? If not they need to address in future directions.
3)
As is now well known, tumors grow and evolve through a constant crosstalk with the surrounding microenvironment, and emerging evidence indicates that angiogenesis and immunosuppression frequently occur simultaneously in response to this crosstalk.
4) Accordingly, strategies combining anti-angiogenic therapy and immunotherapy seem to have the potential to tip the balance of the tumor microenvironment and improve treatment response.
5) In the frame of this thinking, Although decision making strategy based on clinico-histopathological criteria is well established, renal cell carcinoma (RCC) represents a spectrum of biological ecosystems characterized by distinct genetic and molecular alterations, diverse clinical courses and potential specific therapeutic vulnerabilities. Given the plethora of drugs available, the subtype-tailored treatment to RCC subtype holds the potential to improve patient outcome, shrinking treatment-related morbidity and cost. The emerging knowledge of the molecular taxonomy of RCC is evolving, whilst the antiangiogenic and immunotherapy landscape maintains and reinforces their potential. Although several prognostic factors of survival in patients with RCC have been described, no reliable predictive biomarkers of treatment individual sensitivity or resistance have been identified (please refer to PMID: 32456352 and PMID: 32064051 and expand the introduction/discussion sections)
6) The authors need to highlight what new information the review is providing to enhance the research in progress.
7)the authors (also inspired by the suggested references, could came up with a figure a table and/or a graphical abstract summarizing their findings
Minor
- A native speaker revision can be beneficial
Author Response
REVIEWER 2
The authors uncovered microenvironment as a convenient target for pharmacological action Major:
1) The rationale of why the authors came up with this review.
2) What is the information that is not exactly available that motivated the authors to come up with this information.
3)What are the current caveats and how do the authors highlight the current research in answering them? If not they need to address in future directions.
4) As is now well known, tumors grow and evolve through a constant crosstalk with the surrounding microenvironment, and emerging evidence indicates that angiogenesis and immunosuppression frequently occur simultaneously in response to this crosstalk.
As for point 1) the rationale can be summarized into the observation ‘The rise of off-target therapeutic agents is a clear indication of the relevance of microenvironment as a proper focus point of therapy’. An off-target therapy encompasses the recognition of a non-specific (and possibly non-receptor based) way of action of a drug. The same therapy acts on apparently very dissimilar pathological states, thus its ‘target’ must be located in a much more ‘general’ organization layer than the molecular one. To better clarify this point, we changed the title adding the specification ‘chemico-physical’ to microenvironment that is reminiscent of a ‘field effect’ of a drug. As for point 2), it is much easier to underline what is the information that is now available (that in any case is very far to be exact) that the contrary. We simply know (by a large body of experience) that drugs action is very context dependent and we give to this context the generic name of ‘microenvironment’. We can think to microenvironment using the century’s old and glorious name of ‘field’. We can think at an electric field that impinges on an embedded charged particle that in turn modifies the field, by substituting charged particle with cell (tissue) and electric field as the combination of chemical species and physical forces surrounding it. This minimalistic definition is, at least in our opinion, very important in order not to make the message too vague, and we can measure the field only with a very limited number of probes like pH or stiffness of the tissues. This paucity of information is at the basis of the answer to point 3) that is to look-up over the pure receptorial largely deterministic mechanisms. This attitude encompasses directions of research as diverse as mechanobiology, non-linear thermodynamics, exploitation of inorganic therapeutic agents, nanomedicine…
There is concordant scientific evidence that the acidic microenvironment negatively influences both the action of chemical drugs and the immune reaction against tumors as well. As we report and comment into the submitted manuscript there are two reports using two different anti-acidic approaches that improve both the efficacy of an adoptive immune therapy and the natural immune reaction against tumors, by using either proton pump inhibitors (Calcinotto Cancer Res 2012) or sodium bicarbonate (Pilon-Thomas S et al Cancer Res 2016). Thus, with this article we want first to emphasize how is crucial to buffer the acidity of tumor microenvironment in order to allow a more efficient chemical drug and immune therapies efficacy. By analogy with tumors there many disease that are characterized by microenvironmental acidity these includes diabetes and infectious disorders, suggesting that an off targetting of Antiacidic compounds may be broadly useful in a new therapeutic strategy against a variety of diseases.
4) Accordingly, strategies combining anti-angiogenic therapy and immunotherapy seem to have the potential to tip the balance of the tumor microenvironment and improve treatment response.
It is not by chance that we insert the ‘chemico-physical’ nature of our definition of microenvironment. The focus is not toward the cross-talk but toward field effects that escape the classical molecular paradigms. In this respect, even if angiogenic and immune therapies are of sure interest they do not fall in the focus of this manuscript.
Last but not least malignant tumors grow well in the absence of blood, in the hypoxic conditions and in a low nutrient supply, and the anti-angiogenetic approach has led to selection of very malignant cells, in fact living with low blood supply (Pillai et al..).
Going to point 5), the reviewer states:
Although decision-making strategy based on clinico-histopathological criteria is well established, renal cell carcinoma (RCC) represents a spectrum of biological ecosystems characterized by distinct genetic and molecular alterations, diverse clinical courses and potential specific therapeutic vulnerabilities. Given the plethora of drugs available, the subtype-tailored treatment to RCC subtype holds the potential to improve patient outcome, shrinking treatment-related morbidity and cost. The emerging knowledge of the molecular taxonomy of RCC is evolving, whilst the antiangiogenic and immunotherapy landscape maintains and reinforces their potential. Although several prognostic factors of survival in patients with RCC have been described, no reliable predictive biomarkers of treatment individual sensitivity or resistance have been identified (please refer to PMID: 32456352 and PMID: 32064051 and expand the introduction/discussion sections).
While we substantially agree with the reviewer considerations, we must stress this manuscript is not a review but a perspective suggesting the possibility of considering the CHEMICO-PHYSICAL microenvironment as a proper therapeutic target. As a general thought when you have a plethora of drugs available and the tumor patients continue to die at least a doubt on the way the current anti-tumor drugs have been designed should generate in our minds. In fact, the endpoint of this perspective article is to suggest scientists to sit down and think, and as an example of a new strategy we suggest to think to the very hostile microenvironment of tumors as a new target. As a practical example we provide proton pump inhibitors that while being anti-acidic drugs have shown a potent anti-tumor effect and to increase the effectiveness of a panel of anti-tumor drugs..
6) The authors need to highlight what new information the review is providing to enhance the research in progress.
The new information implies that pharmacological research should move from cellular targets to microenvironmental targets, in terms of physical-chemical features. Again we provide the example of proton pump inhibitors that have shown to be effective against vacuolar ATPases while their main targets are the gastric H+/K+/ATPases, but also to have other off-targets including some nervous system associated receptors. Thus, the goal of our article is mainly methodological and with new perspective for both pharmacological investigation and medicine. This has been further discussed and emphasized in the revised version
7)the authors (also inspired by the suggested references, could came up with a figure a table and/or a graphical abstract summarizing their findings
We newly included a figure showing the pH gradients in normal and pathological conditions ,a table summarizing literature data on the importance of chemical-physical conditions in generating diseases, with including additional 102 references
Minor
A native speaker revision can be beneficia.
Thanks it has been revised
Round 2
Reviewer 1 Report
Even though the authors gave appropriate responses to most of the inquires, the English language still needs to be improved. Please do not ask for the 'aid of a native speaker'. That is not what was required. It was required to submit the manuscript into a professional editing service, and to show the proof of it. The manuscript must be evaluated by a professional English editing service. The new version of the manuscript provided by the authors for this round of review still has several flaws with the language. Examples include, but definitely are not limited to:
1. (Line 160) "Actually, some strategies have been exploited to exploit...".
2. (Line 113) Double '.' at the end of the paragraph.
3. The way the whole paragraph from lines 114 to 118 is written is confusing due to the lack of punctuation marks.
4. In the text listing the approaches for targeting tumor acidosis (lines 148-150), sometimes the authors use ';' when separating the numbers in the list, and sometimes they use ',' instead.
5. (Line 162) The fact that for the authors and their native speaker, 'sodium bicarbonate' should be written as 'Sodium Bicarbonate'. This is the same for all the drugs that were indicated in the previous round of the review, and which are seen in lines 247-251.
6. (Line 204) Do 'anti-biograms' exist? Or should they be 'antibiograms'?
7. (Line 227) 'Targetting'.
8. (Line 236) Instead of 'clones 'selection', shouldn't it be 'clones' selection'?
9. Does 'pantpprazole' exist for the native speaker of the authors?
10. What is a 'plasmamembrane'? This is seen in lines 294-295.
11. The way the whole paragraph from lines 303 to 307 is written is confusing due to the linkers used.
12. In line 307, should it be 'summarize' or instead 'summarizes'?
12. Since the 'NBC' acronym was already explained in line 235, is it any particular reason to repeat it in line 254? It is the same for 'PPI' seen both at lines 165 and 282.
Additionally, in several sections of the text the authors have added an extra an unnecessary space between words, such as:
1. Line 44 (between 'repurposing' and 'stems').
2. Line 68 (between 'notion' and ',').
3. Line 71 (between 'exhaustively' and 'take).
4. Line 73 (between 'by' and 'pharmacological').
5. Line 94 (between 'systems' and '.').
6. Line 94 (between 'organization' and ',').
7. Line 102 (between 'the' and 'initial').
8. Line 132 (between '.' and 'This').
9. Line 134 (between '.' and 'Indeed').
10. Line 135 (between '.' and 'The').
11. Line 139 (between '.' and 'Independently').
12. Line 141 (between '.' and 'Rather').
13. Line 146 (between '.' and 'Thus').
14. Line 161 (between '[26,33]' and '.').
15. Line 236 (between '.' and 'In').
16. Line 279 (between ';' and '(ii)').
17. Line 281 (between 'represent' and 'a').
18. Line 289 (between '[34]' and 'In').
19. Line 300 (between 'both' and 'repurposed').
20. Line 304 (between 'support' and ',').
21. Line 304 (between 'tumors' and ',').
On the other hand, it is not clear what reference software the authors used, as the references seen in lines 293 and 294 have a different style than those seen in the rest of the text.
Finally, the authors replied that they do not want to add immunology- or exosome-related topics in their perspective manuscript. However, they do briefly talk about exosomes (line 295) and immunity (lines 296-306). It is requested that the authors either discuss in a proper way both topics, or that they do not include them at all if they want to continue with their vision of the microenvironment as a field.
Author Response
Rev 1
Comments and Suggestions for Authors
Even though the authors gave appropriate responses to most of the inquires, the English language still needs to be improved. Please do not ask for the 'aid of a native speaker'. That is not what was required. It was required to submit the manuscript into a professional editing service, and to show the proof of it. The manuscript must be evaluated by a professional English editing service. The new version of the manuscript provided by the authors for this round of review still has several flaws with the language. Examples include, but definitely are not limited to:
- (Line 160) "Actually, some strategies have been exploited to exploit...".
- (Line 113) Double '.' at the end of the paragraph.
- The way the whole paragraph from lines 114 to 118 is written is confusing due to the lack of punctuation marks.
- In the text listing the approaches for targeting tumor acidosis (lines 148-150), sometimes the authors use ';' when separating the numbers in the list, and sometimes they use ',' instead.
- (Line 162) The fact that for the authors and their native speaker, 'sodium bicarbonate' should be written as 'Sodium Bicarbonate'. This is the same for all the drugs that were indicated in the previous round of the review, and which are seen in lines 247-251.
- (Line 204) Do 'anti-biograms' exist? Or should they be 'antibiograms'?
- (Line 227) 'Targetting'.
- (Line 236) Instead of 'clones 'selection', shouldn't it be 'clones' selection'?
- Does 'pantpprazole' exist for the native speaker of the authors?
- What is a 'plasmamembrane'? This is seen in lines 294-295.
- The way the whole paragraph from lines 303 to 307 is written is confusing due to the linkers used.
- In line 307, should it be 'summarize' or instead 'summarizes'?
- Since the 'NBC' acronym was already explained in line 235, is it any particular reason to repeat it in line 254? It is the same for 'PPI' seen both at lines 165 and 282.
Additionally, in several sections of the text the authors have added an extra an unnecessary space between words, such as:
- Line 44 (between 'repurposing' and 'stems').
- Line 68 (between 'notion' and ',').
- Line 71 (between 'exhaustively' and 'take).
- Line 73 (between 'by' and 'pharmacological').
- Line 94 (between 'systems' and '.').
- Line 94 (between 'organization' and ',').
- Line 102 (between 'the' and 'initial').
- Line 132 (between '.' and 'This').
- Line 134 (between '.' and 'Indeed').
- Line 135 (between '.' and 'The').
- Line 139 (between '.' and 'Independently').
- Line 141 (between '.' and 'Rather').
- Line 146 (between '.' and 'Thus').
- Line 161 (between '[26,33]' and '.').
- Line 236 (between '.' and 'In').
- Line 279 (between ';' and '(ii)').
- Line 281 (between 'represent' and 'a').
- Line 289 (between '[34]' and 'In').
- Line 300 (between 'both' and 'repurposed').
- Line 304 (between 'support' and ',').
- Line 304 (between 'tumors' and ',').
On the other hand, it is not clear what reference software the authors used, as the references seen in lines 293 and 294 have a different style than those seen in the rest of the text.
Finally, the authors replied that they do not want to add immunology- or exosome-related topics in their perspective manuscript. However, they do briefly talk about exosomes (line 295) and immunity (lines 296-306). It is requested that the authors either discuss in a proper way both topics, or that they do not include them at all if they want to continue with their vision of the microenvironment as a field.
- WE HAVE AMENDED OR CHANGED ALL THE PARTS SUGGESTED BY THE REVIEWER AND ASKED TO A PROFESSIONAL ENGLISH EDITING SERVICE TO REVISE OUR MANUSCRIPT. THE CERTIFICATE HAS BEEN INCLUDED AS APPROPRIATE
- AS FOR THE EXOSOMES AND IMMUNITY ISSUES, WHILE WE CAN AGREE WITH THE REVIEWER THAT THEY SHOULD BE PRESENTED MORE EXTENSIVELY, AS WE HAVE CONTINUOSLY REPEATED ALL ALONG THE MANUSCRIPT OUR AIM WAS AND IS TO DEAL WITH THE EVIDENCE THAT IS MORE THE MICROENVIRONMENT AND NOT CELLS AND OTHER FACTORS (E.G. EXOSOMES) THAT MAY INDUCE A CHANGE THAT MAY OR MAY NOT LEAD TO THE DEVELOPMENT OF A DISEASE (E.G. CANCER). THEREFORE THERAPIES SHOULD BE DIRECTED TO THE MICROENVIRONMENT RATHER THAN TO THE CELLS . ONE KEY PAPER WAS LOGOZZI ET AL CANCERS 2018 (REF 12 OF THE MANUSCRIPT) IN WHICH THE SAME HUMAN CANCER CELLS OF DIFFERENT ORIGINS ARE CULTURED AT EATHER ACIDIC OR ALKALINE CONDITIONS AND PROGRESSIVELY BROUGHT TO THE OPPOSITE pH CONDITION , SHOWING THAT EXOSOMES RELEASE WERE ALWAYS SIGNIFICANTLY HIGHER AT LOW pH (I.E. THE pH OF THE TUMORS) DECREASING WITH THE INCREASE OF THE pH.
THUS WE WANTED AND WANT TO FOCUS ON THIS ISSUE NOT DEDICATING TOO MUCH TO OTHER ISSUES THAT ARE VERY IMPORTANT BUT FOR THIS REASON DESERVING A HUGE SPACE TO BE TREATED IN THE PROPER WAY AND LEADING ASTRAY FROM THE MAIN FOCUS OF THIS PERSPECTIVE ARTICLE
HOWEVER, WE REALLY WANT TO THANK THE REVIEWER FOR HER/HIS VERY CAREFULLY READING OF OUR MANUSCRIPT

Reviewer 2 Report
The authors have clarified several of the questions I raised in my previous review. When it comes to inhibiting gastric acid secretion, PPIs are the most effective. These medications are commonly prescribed today and can be used for a variety of purposes. According to the Cheung and Brusselaers studies, chronic use is associated with an increased risk of gastric cancer. Both dose and duration contribute to this increase in risk. Additionally, the risk increases across all ages and both sexes. Many people use PPIs without any indication for their use, so it is recommended to limit their use to those with clearly defined indications and to assess whether their indications are still valid. Those with gastric risk factors for cancer should also be treated promptly, such as chronic H. pylori infection and peptic ulcers. As well as existing confounders, future studies will be needed to fully explain the effects of chronic PPI use. In addition to defining the maximum duration of use of PPIs that is associated with the lowest risk of gastric cancer, future studies are also needed to answer the other above-mentioned questions. Last but not least, I would like to thank all the researchers who have worked on this topic and encourage others to continue researching this topic until we have all the answers. While acknowledging the author efforts to answer this reviewr's criticisms, Most of the major problems have not been addressed by this revision. Nevertheless, as I stated in my previous review, I deem it unlikely that all those (partially minor) issues can be solved merely by a few added sentences: is still this reviewer's opinion that the manuscript would have benefited from an introduction/discussion sections expansion (as already suggested), nonetheless, the reviewing process, is also subjective: the last decision is for the Editor.
The English is still subpar.
Author Response
REV 2
Comments and Suggestions for Authors
The authors have clarified several of the questions I raised in my previous review. When it comes to inhibiting gastric acid secretion, PPIs are the most effective. These medications are commonly prescribed today and can be used for a variety of purposes. According to the Cheung and Brusselaers studies, chronic use is associated with an increased risk of gastric cancer. Both dose and duration contribute to this increase in risk. Additionally, the risk increases across all ages and both sexes. Many people use PPIs without any indication for their use, so it is recommended to limit their use to those with clearly defined indications and to assess whether their indications are still valid. Those with gastric risk factors for cancer should also be treated promptly, such as chronic H. pylori infection and peptic ulcers. As well as existing confounders, future studies will be needed to fully explain the effects of chronic PPI use. In addition to defining the maximum duration of use of PPIs that is associated with the lowest risk of gastric cancer, future studies are also needed to answer the other above-mentioned questions. Last but not least, I would like to thank all the researchers who have worked on this topic and encourage others to continue researching this topic until we have all the answers. While acknowledging the author efforts to answer this reviewr's criticisms, Most of the major problems have not been addressed by this revision. Nevertheless, as I stated in my previous review, I deem it unlikely that all those (partially minor) issues can be solved merely by a few added sentences: is still this reviewer's opinion that the manuscript would have benefited from an introduction/discussion sections expansion (as already suggested), nonetheless, the reviewing process, is also subjective: the last decision is for the Editor.
The English is still subpar.
WE THANK THE REVIEWER FOR HER/HIS READING OF OUR MANUSCRIPT AND FOR THE FRUITFUL COMMENTS.
HOWEVER, AS FAR AS PPI AND THEIR SIDE EFFECTS ARE CONCERNED WE WOULD LIKE TO EMPHASIZE THE FOLLOWING ISSUES:
1.OUR ARTICLE PROPOSES THE USE OF PPI IN TREATING SERIOUS DISEASES SUCH AS CANCER, WHERE CURRENTLY ARE USED DRUGS THAT ARE POISONS AT VERY HIGH DOSAGES THAT TOO OFTEN INDUCE MORE SIDE EFFECTS THAN CURE. WHILE REPORTS ON SIDE EFFECTS RELATED TO PPI HAVE BEEN PUBLISHED WE GUESS THAT TALKING ABOUT CANCER AND ALSO MANY OTHER CHRONIC DISEASES IT IS NOT THE CASE TO TALK ABOUT SIDE EFFECTS, RATHER ABOUT EFFICACY.
2.WE AGREE THAT THE UNCONTROLLED USE OF PPI BY HUNDRED MILLIONS OF PEOPLE WORLDWIDE MAY BE A PROBLEM, AND IT IS A DUTY OF SCIENTISTS AND MEDICAL DOCTORS TO INFORM USERS AND ABUSERS ON THE POTENTIAL SIDE EFFECTS. HOWEVER, DUE TO THE UNBELIEVABLE AMOUNT OF PEOPLE ASSUMING PPI WORLDWIDE THERE ARE NOT TOO MANY OFFICIAL REPORTS ON DIFFUSED PPI-RELATED SIDE EFFECTS, SUGGESTING THAT , EVEN POSSIBLE, PPI-RELATED SIDE EFFECTS ARE NOT SO FREQUENT.
3.THERE ARE TWO RECENT PAPERS SUGGESTING THAT PEOPLE ASSUMING ANTIACIDIC DRUGS INCLUDING PPI SHOW REDUCED INCIDENCE OF CANCER OR INCREASED OVERALL SURVIVAL EVEN IF AFFECTED BY VERY AGGRESSIVE CANCERS, BOTH REPORTED BELOW AND INCLUDED INTO THE REVISED MANUSCRIPT
Go S, Lee DY, Choi WI, Jeong J. Association between use of antacid medications (proton pump inhibitors and histamine-2 receptor antagonists) and the incidence of lung cancer: A population-based cohort analysis. Medicine (Baltimore). 2022 Sep 9;101(36):e30399. doi: 10.1097/MD.0000000000030399. PMID: 36086741.
Han JH, Jeong SH, Yuk HD, Jeong CW, Kwak C, Ku JH. Acidic urine is associated with poor prognosis in patients with bladder cancer undergoing radical cystectomy. Front Oncol. 2022 Aug 26;12:964571. doi: 10.3389/fonc.2022.964571. PMID: 36091123; PMCID: PMC9459327.
- WE DON’T WANT HERE TO EXCLUSIVELY SUPPORT THE USE OF PPI IN CANCER TREATMENT – RATHER WE WOULD LIKE TO EMPHASIZE THE IMPORTANCE OF THINKING ABOUT A NEW OPPORTUNITY FOR DRUG DISCOVERY THAT IS TO USE THE PHYSICAL-CHEMICAL PHENOTYPE OF DISEASE MICROENVIRONMENT AS A REAL TARGET FOR FUTURE MOLECULES – OF COURSE PROTON PUMP INHIBITORS MAY REPRESENT A SOURCE OF INSPIRATION BEING A CLASS OF PRODRUGS THAT ONLY WHEN PROTONATED BECOME THE ACTIVE MOLECULE (TETRACYCLIC SULFENAMIDE).
AS WRITTEN FOR THE REV 1 THE ENGLISH UNDERWENT PROFESSIONAL ENGLISH EDITING AND THE CERTIFICATE IS PROVIDED

Round 3
Reviewer 1 Report
I have nothing else to comment.